# Employing Gamma-Ray-Modified Carbon Quantum Dots to Combat a Wide Range of Bacteria

**DOI:** 10.3390/antibiotics12050919

**Published:** 2023-05-17

**Authors:** Zoran M. Marković, Aleksandra S. Mišović, Danica Z. Zmejkoski, Nemanja M. Zdravković, Janez Kovač, Danica V. Bajuk-Bogdanović, Dušan D. Milivojević, Marija M. Mojsin, Milena J. Stevanović, Vladimir B. Pavlović, Biljana M. Todorović Marković

**Affiliations:** 1Vinča Institute of Nuclear Sciences, National Institute of the Republic of Serbia, University of Belgrade, Mike Alasa 12-14, 11001 Belgrade, Serbia; zoranmarkovic@vin.bg.ac.rs (Z.M.M.); aleksandra.misovic@vin.bg.ac.rs (A.S.M.); danica@vinca.rs (D.Z.Z.); dusanm@vinca.rs (D.D.M.); 2Scientific Veterinary Institute of Serbia, Janisa Janulisa 14, 11107 Belgrade, Serbia; nemanja.zdravkovic@nivs.rs; 3Jozef Stefan Institute, Department of Surface Engineering—F4, Jamova cesta 39, 1000 Ljubljana, Slovenia; janez.kovac@ijs.si; 4Faculty of Physical Chemistry, University of Belgrade, Studentski trg 12-16, 11158 Belgrade, Serbia; danabb@ffh.bg.ac.rs; 5Institute of Molecular Genetics and Genetic Engineering, University of Belgrade, Vojvode Stepe 444a, 11042 Belgrade, Serbia; mojsin@imgge.bg.ac.rs (M.M.M.); milenastevanovic@imgge.bg.ac.rs (M.J.S.); 6Faculty of Biology, University of Belgrade, Studentski trg 16, 11000 Belgrade, Serbia; 7Serbian Academy of Sciences and Arts, Knez Mihailova 35, 11000 Belgrade, Serbia; 8Faculty of Agriculture, University of Belgrade, Nemanjina 6, Zemun, 11080 Belgrade, Serbia; vlaver@agrif.bg.ac.rs

**Keywords:** carbon quantum dots, gamma rays, antibacterial activity, cellular uptake

## Abstract

Nowadays, it is a great challenge to develop new medicines for treating various infectious diseases. The treatment of these diseases is of utmost interest to further prevent the development of multi-drug resistance in different pathogens. Carbon quantum dots, as a new member of the carbon nanomaterials family, can potentially be used as a highly promising visible-light-triggered antibacterial agent. In this work, the results of antibacterial and cytotoxic activities of gamma-ray-irradiated carbon quantum dots are presented. Carbon quantum dots (CQDs) were synthesized from citric acid by a pyrolysis procedure and irradiated by gamma rays at different doses (25, 50, 100 and 200 kGy). Structure, chemical composition and optical properties were investigated by atomic force microscopy, transmission electron microscopy, X-ray photoelectron spectroscopy, Fourier transform infrared spectroscopy, Raman spectroscopy, UV-Vis spectrometry and photoluminescence. Structural analysis showed that CQDs have a spherical-like shape and dose-dependent average diameters and heights. Antibacterial tests showed that all irradiated dots had antibacterial activity but CQDs irradiated with dose of 100 kGy had antibacterial activity against all seven pathogen-reference bacterial strains. Gamma-ray-modified CQDs did not show any cytotoxicity toward human fetal-originated MRC-5 cells. Moreover, fluorescence microscopy showed excellent cellular uptake of CQDs irradiated with doses of 25 and 200 kGy into MRC-5 cells.

## 1. Introduction

Bacteria colonization on different surfaces is currently a big problem because bacteria are very easily transferred from infected surfaces (glass, wood, plastics) to humans [1,2,3]. Another major problem is bacteria developing a resistance to existing and applied antibiotics. Thus, it is very important to develop highly effective medicines for the treatment of different bacteria strains. There are few therapeutic strategies to combat enhanced antibiotic resistance and one of them is the use of photosensitizers as medicines [4]. Different nanoparticles (nitric-oxide-releasing nanoparticles, chitosan-containing nanoparticles and metal-containing nanoparticles) can overcome various drug resistance mechanisms, including decreased uptake, increased efflux of a drug from the pathogen cell, biofilm formation and intracellular bacteria, by using multiple mechanisms simultaneously to combat pathogens [5]. 

Antimicrobial photodynamic therapy is a type of photodynamic therapy in which materials called photosensitizers produce reactive oxygen species under light at a certain wavelength and this eradicates different types of pathogens (bacteria, viruses or fungi) [6,7,8]. CQDs, as one of the highly potent photosensitizers, are zero-dimensional carbon-based materials with an average diameter up to 20 nm, with good chemical and thermal stability and resistance to photo-bleaching [9]. The carbon networks of CQDs are predominantly composed of amorphous and crystalline cores [10]. CQDs are photoactive, i.e., they emit light which can be tuned from blue to red depending on the excitation wavelength. Their photoluminescence can be shifted in different parts of electromagnetic spectra by various modifications of CQDs, for example, by doping or by gamma or ion irradiation. By surface functionalization and passivation with different materials, the physical and chemical properties of CQDs can be modified significantly. Doping of CQDs with different atoms (F, Cl, N, S, B) can very significantly affect the properties of CQDs (structural, photoluminescent, antibacterial) [11]. 

Many researchers have prepared different types of CQDs (doped or modified by various procedures) and investigated their antibacterial and cytotoxic activities [12,13,14,15,16,17,18]. Du et al. showed that there was a correlation between the size and antibacterial activity of CQDs prepared by an electrochemical oxidation method using L-ascorbic acid as the raw material [19]. Chai et al. reported that phosphorus-doped CQDs prepared hydrothermally from m-aminophenol and phosphoric acid showed effective antibacterial activity against *Escherichia coli* (*E. coli*) and *Staphylococcus aureus* (*S. aureus*) [20]. Marković et al. noticed that CQDs prepared from o-Phenylenediamine had very poor antibacterial potential due to a lack of reaction centers [21]. In addition, Bianco et al. claimed recently that pyridine nitrogen can be a reactive center and activates a reactive center at the adjacent carbon atoms in the functionalized C-N bonds for additional post reactions such as oxidation [22]. 

In this manuscript, we have investigated the antibacterial and cytotoxic activities of a gamma-ray-modified CQD structure against seven different bacteria strains. To the best of our knowledge, data about irradiated CQDs’ antimicrobial activities are sparse. Gamma rays are electromagnetic waves with high energy and different modifications of materials can be performed by using this type of irradiation [23]. Apart from structural modification of prepared CQDs (particle size, chemical composition, photoluminescence and pro-oxidant activity), we investigated how different doses of gamma rays affected the antibacterial and cytotoxic activities against various bacteria strains—*S. aureus*, *Methicillin-resistant S. aureus (MRSA)*, *E. coli, Klebsiella pneumoniae (K. pneumoniae)*, *Proteus mirabilis (P. mirabilis)* and *Pseudomonas aeruginosa (P. aeruginosa)*—and MRC-5 cells, respectively. 

## 2. Results

### 2.1. Surface Morphology

Transmission electron microscopy (TEM) and atomic force microscopy (AFM) were used to visualize morphology—the size and height distribution of CQDs gamma irradiated with doses of 25, 50, 100 and 200 kGy. The irradiated samples were designated as follows: CQD_25, CQD_50, CQD_100 and CQD_200, respectively. The TEM micrographs are presented in Appendix A whereas top view AFM images with particle size and height distributions are shown in Figure 1a–d. Statistical analysis has been conducted on more than 20 AFM images of CQDs with total particles numbering larger than 500. As can be seen from Figure 1a, more than 40% of CQD_25 dots have a diameter of 15 nm whereas the average diameter is 26.4 nm. The average height of these dots is 5.5 nm. Figure 1b shows that about 61% of CQD_50 particles have a diameter of 15 nm whereas the average diameter is 19.4 nm. The average height of CQD_50 particles is 6.3 nm. Of the CQD_100 nanoparticles, 65% have a diameter of 15 nm, whereas the average diameter is 19.0 nm. The average height of these dots is 11.8 nm, as shown in Figure 1c. Of the CQD_200 nanoparticles, 58% have a diameter of 15 nm, whereas their average diameter is 22.4 nm. The average height of CQD_200 particles is 6.9 nm, as shown in Figure 1d. Based on statistical analysis, we conclude that gamma irradiation of CQDs affects their diameters and heights: with an increase in irradiation dose from 25 kGy to 100 kGy there is an average diameter decrease, and at a dose of 200 KGy the average diameter increases compared with doses of 50 and 100 kGy. In the case of average height there is dose dependence as well: with increase in irradiation, average height is increased with dosage up to 200 kGy. At this level of irradiation, average height decreases to 6.9 nm.

### 2.2. Chemical Composition

To determine the chemical composition of gamma-irradiated CQD samples, X-ray photoelectron spectroscopy (XPS) and Fourier transform infrared spectroscopy (FTIR measurements were conducted. Appendix A presents a survey spectrum of CQD_0 (non-irradiated sample). Similar XPS survey spectra were obtained for other investigated samples. Appendix A and Table 1 and Appendix A show surface composition in at.% measured from the XPS spectra after irradiation by gamma rays at different doses. Carbon and oxygen are mainly present on the surface of all investigated samples. After irradiation, a small decrease in O content is observed (at doses of 50 and 100 kGy compared with the non-irradiated sample) as well as a decrease in N atoms. The presence of nitrogen is probably due to the nitrogen adsorbed on the CQD_0 sample surface.

XPS spectra of carbon C 1s and oxygen O 1s were fitted with different sub-peaks to identify different species present on the surface of samples. The C 1s spectrum was fitted with four peaks C1-C4 representing different bonds as follows: peak C1 at ca. 284.8 eV assigned to C-C/C-H bonds, peak C2 at ca. 286.0 eV assigned to C-O/C-OH/C-O-C bonds, peak C3 assigned to ca. 288.1 eV to C=O/O-C-O/CO_3_ bonds, peak C4 at ca. 288.8 eV assigned to O=C-O [24,25]. Peaks O1 and O2 were identified in the O 1s spectra, O1 at ca. 531.2 eV representing O=C bonds and O2 at ca. 532.5 eV representing O-C bonds [24,25]. Figure 2a–e show fitted C 1s spectra and Appendix A show fitted O 1s spectra. Table 1 presents the relative concentration of these peaks, normalized to 100%. The O/C ratios appear to be dependent on the applied irradiation dose (Figure 2f).

The main peak in the C 1s spectra is C1 related to graphite-lattice C-C/C-H bonds (70–80%). It may be partially also related to a contamination layer. This peak decreases after the first treatment (25 kGy), probably due to the removal of the contamination layer. It increases after applying a dose of 50 kGy; it is the most intense after the dose of 100 kGy, and then it is slightly decreased. The variation of the relatively small peaks C2, C3 and C4 is similar. They are present in the CQDs_0 sample, persist after applying doses of 25 kGy and 50 kGy and then became smaller for large-dose irradiation. There is a difference for the C3 peak with respect to C2 and C4, which increases greatly after the largest dose is applied. 

The O2 peak related to O-C bonds in the oxygen O 1s spectra is already present in the CQDs_0 sample; it reaches the maximum after a dose of 25 kGy and then decreases for larger doses. The opposite is observed for the O1 peak in the O 1s oxygen spectra, possibly representing O=C bonds. Its relative intensity decreases after a dose of 25 kGy and then it starts to increase for larger doses. After the largest dose (200 kGy) it is the main O1 peak in the O 1s spectrum. 

Our XPS results show the changes in the surface chemistry of irradiated CQDs, such as a decrease in O concentration after the first two doses are applied, but an increase in total O after the largest dose, removal of N and changes in the type of chemical bonding of C and O. 

Appendix A shows FTIR spectra of all samples irradiated at different doses. In the CQD_25 sample, the following peaks could be identified: peaks at 3327 and 2662 cm^−^^1^ (O-H stretching vibrations), peak at 2980 cm^−^^1^ stemming from C-H stretching vibrations, peak at 1754 cm^−^^1^ representing C=O vibrations; peak at 1563 cm^−^^1^ due to C=C stretching vibrations; peak at 1386 cm^−^^1^ stemming from O-H bending vibrations and peak at 1058 cm^−^^1^ due to C-O vibrations [26,27]. In the CQD_50 and CQD_100 samples the same bands as in the CQD_25 sample, up-shifted to 3–5 cm^−^^1^, can be detected. In the CQD_200 sample, a peak at 777 cm^−^^1^ is due to C-H bending vibrations, and the peaks at 1321 and 1396 cm^−^^1^ are due to O-H bending vibrations; the peak at 1573 cm^−^^1^ can be attributed to C=C stretching vibrations whereas the peak at 1639 cm^−^^1^ is due to C-O stretching vibrations. The peaks at 2992 and 3369 cm^−^^1^ are due to C-H stretching vibrations and O-H stretching vibrations, respectively. Compared with the CQD_0 sample, all peaks except the peak at 777.5 cm^−^^1^ appear in all the irradiated samples. 

### 2.3. Raman Spectroscopy

Appendix A presents Raman spectra of CQD_0, CQD_25, CQD_50, CQD_100 and CQD_200 samples. Two prominent peaks can be identified in the Raman spectra of all samples: the first is at 1302 cm^−^^1^ (D band) and the second is at 1598 cm^−^^1^ (G band). It is well known that the appearance of the G band is due to double degenerate phonon mode (E_2g_ symmetry) at the Brillouin zone center [28] whereas the existence of the D band is from intervalley double resonance processes, because of the short-range disorder potentials (e.g., adatoms, vacancies and defects) [29]. All recorded Raman spectra (D and G bands) were fitted by Lorentzian functions. Table 2 and Appendix A present the fitted Raman spectra of the CQD_0 sample and irradiated samples CQD_25, CQD_50, CQD_100 and CQD_200. The G peaks of the samples were fitted by three Lorentzian peaks (1580, 1590 and 1610 cm^−^^1^). These peaks are denoted as G11, G12 and G2. 

The lower-frequency component attributed to the carbon vibrations in the interior of the graphite layers (a frequency range from 1580 to 1590 cm^−^^1^—G1) is split into a doublet (G11 and G12) in the CQD_100 and CQD_200 samples. The upper-frequency component G2 is due to carbon vibrations between the bound graphite layers, and adjacent to functional groups (a frequency range from 1600 to 1630 cm^−^^1^—G2) [30]. The I_D_/I_G_ ratio is dose dependent. As can be seen from Table 2, the I_D_/I_G_ ratio increases with applied dose (from 1.6 to 2.1).

### 2.4. UV-Vis and PL Measurements

The UV-Vis spectra of all samples are presented in Appendix A. It can be seen from this figure that the CQD_0 sample has two absorption bands: one at 246 nm and the other at 291 nm, whereas the CQD_200 sample also has two downshifted absorption bands: one at 235 nm and the other at 266 nm. The band at 246 nm (235 nm) represents the π–π* transition of C=C, whereas the other bands correspond to the n–π* transition of C=O [31]. 

Photoluminescence (PL) of carbon dots can be caused by different mechanisms: a quantum core effect due to the π -conjugated domains in the carbon dot core, surface state PL due to doping by various heteroatoms, hybridization of the carbon honeycomb and surface functional groups, and finally PL caused by fluorophores and crosslink-enhanced PL emission [32]. Figure 3 presents the PL spectra of CQD_0, CQD_25, CQD_50, CQD_100 and CQD_200 samples. There is PL excitation–emission dependence for all investigated CQD samples with the highest PL intensity under an excitation wavelength of 350 nm (Figure 3), with the PL emission up-shifted from 484.65 nm (CQD_0) to 498.46 nm (CQD_200). PL spectra of all samples were fitted to two Gaussian peaks (P1 and P2) and the results are shown in Table 3. The P1 peak represents the core emission whereas the P2 peak originates from defects on the core edges [33]. The P peak is the maxima intensity peak at certain irradiation doses. 

There are up-shifts of the P1, P2 and P peaks depending of the applied irradiation dose based on data in Table 3. The S1/S2 ratio represents integrated surfaces under the P1 and P2 peaks, which decreases with the applied dose, i.e., raising the irradiation dose increases the distribution of defects on the core edges.

Table 4 presents the quantum yield (QY) of fluorescence of all samples. It can be seen from this table that there is a decreasing trend of QY with the increase in irradiation dose.

### 2.5. Singlet Oxygen Generation

To determine the singlet oxygen production, electron paramagnetic resonance (EPR) and UV-Vis spectrophotometry were used. The singlet oxygen spin trap (2,2,6,6-tetramethylpiperidine-TEMP) electron paramagnetic resonance intensities of the CQD_25 and CQD_200 samples exposed to blue light (BL) are shown in Appendix A. TEMP molecules rapidly react with ^1^O_2_, and form a stable, EPR-active product, TEMP-^1^O_2_ (TEMPO). Commercially available TEMP spin traps contain a small amount of TEMPO as an impurity, which contributes to a small parasitic signal. Upon exposure to singlet oxygen, the spin trap EPR spectrum intensities significantly increase, revealing singlet oxygen formation. 

The EPR measurements of the CQD_25, CQD_200, and control samples exposed to BL showed a considerable rise in EPR intensity, indicating singlet oxygen formation (Appendix A). These results demonstrate that CQD_25 and CQD_200 samples generate singlet oxygen upon exposure to BL irradiation. 

The same experiment was conducted under ambient light (AL) conditions. The obtained results show that none of the tested samples generated singlet oxygen.

The other method used to determine singlet oxygen production was the measurement of the absorption spectra of 9,10-Anthracenediyl-bis(methylene)dimalonic acid (ABDA)—in different time intervals under AL and BL irradiation, as shown in Figure 4 [34].

All measurements were repeated three times to ensure accuracy. As can be seen from Figure 4a, singlet oxygen production did not take place under AL conditions. However, under BL conditions, there was singlet oxygen production by all irradiated samples. There were no significant differences among levels of singlet oxygen production of the irradiated samples. Gamma-irradiated samples produced smaller amounts of singlet oxygen compared with the CQD_0 sample.

Both methods confirmed that all irradiated samples produced singlet oxygen under BL irradiation. In our previous research we established that CQDs produced from citric acid did not produce any other type of reactive oxygen species (i.e., hydroxyl radicals or even superoxide anions) [35]. Thus, the main mechanism for singlet oxygen production is the energy transfer from the photosensitizer (CQDs) to the O_2_ molecules [36]. 

### 2.6. Antibacterial Activity

Comparative antibacterial properties for a pair of AL- and BL-treated CQD colloids are illustrated in Figure 5a–g. The CQD_25, CQD_50, CQD_100 and CQD_200 samples showed inhibition of bacterial growth. Even though there were 11 twofold dilution steps, high uniformity of OD has been achieved. A remarkable increase in the inhibiting effect has been achieved for CQD_100 colloids in all tested bacteria, and CQD_200 and CQD_25 samples each produced significant differences in two of the seven tested bacteria groups*—K. pneumonie* and *Salmonella typhimurim (S. typhimurim)*; and *S. aureus* and MRSA, respectively. 

Under AL irradiation, the sample CQD_200 showed a significant antibacterial effect against MRSA, *E. coli*, *P. mirabilis* and *P. aeruginosa*. When comparing AL and BL conditions, the sample CQD_100 showed an antibacterial effect against all tested strains with significantly better antibacterial effect when samples were exposed to BL (*p* < 0.0001). Furthermore, against Gram-positive MRSA and *S. aureus*, sample CQD_25 showed a better effect under BL (*p* < 0.01 and *p* < 0.001, respectively). When observing Gram-negative strains, sample CQD_200 showed a significant effect against *K. pneumoniae* and *S. typhimurim* when exposed to BL irradiation.

### 2.7. Morphology of Bacterial Strains

It is very important to visualize bacteria strains after treatment with different types of CQDs under AL and BL conditions. AFM was used to visualize the effect of the CQD_100 sample on the morphology of MRSA under AL and BL conditions. The morphology of MRSA bacteria strains after treatment with the CQD_100 sample under AL and BL irradiation is presented in Figure 6a,b. The average diameter of MRSA after treatment with CQD_100 is 1.88 μm whereas the height is 0.40 μm under AL. 

The average diameter of MRSA after treatment with CQD_100 is 1.22 μm whereas the height is 0.27 μm under BL irradiation, showing the reduction in average bacteria size in diameter to be about 64.9% whereas the height is reduced by about 67.5%.

### 2.8. Cytotoxicity

Cytotoxic effects of CQD_25 and CQD_200 samples were tested against MRC-5 cells using MTT assay after 24 h of treatment with increasing concentration of CQD samples [37,38]. As shown in Figure 7, all samples showed a non-cytotoxic effect and cell viability >80%, regardless of the concentrations of tested samples. The viability range was within 89–101% for all concentrations of all tested samples (Figure 7). 

### 2.9. Fluorescence Microscopy

Internalization of CQD_25 and CQD_200 in MRC-5 cells is presented in Figure 8. MRC-5 cells were treated with CQD_25 or CQD_200 with a concentration of 200 µg/mL for 48 h and cellular uptake was visualized by fluorescent microscopy (Figure 8). As shown, both irradiated CQD samples penetrate into MRC-5 cells marked by red arrows (Figure 8b,c).

## 3. Discussion

The effect of gamma rays on the structural, chemical and photoluminescent properties of CQDs, and further, their antibacterial and cytotoxic potentials, is the main focus of this manuscript. AFM statistical analysis showed that all investigated samples had a sphere-like shape and there was dose dependence regarding dot diameters and heights. XPS and FTIR analyses demonstrated that the O/C ratio depended on the irradiation dose, i.e., there was a rising trend in this ratio. PL measurements showed that CQDs emit light after gamma irradiation and there was excitation–emission dependence of PL. Furthermore, raising the irradiation dose increased the distribution of defects on the core edges.

Further, antibacterial and cytoxic tests were conducted with four gamma-ray-modified CQD samples. All tested samples demonstrated antibacterial activity. A remarkable increase in inhibiting effect was achieved for CQD_100 colloids in all tested bacteria, and CQD_200 and CQD_25 samples each produced significant differences in two of seven tested bacteria groups—*K. pneumonie* and *S. typhimurim*; and *S. aureus* and *MRSA*, respectively. Cytotoxicity tests showed a high level of MRC-5 cell viability. In our previous research, we established that CQD_0 samples had antibacterial activity toward a wide range of different bacteria strains [35]. Namely, the MIC concentrations were in the range of 7.81–125 μg/mL whereas *K. pneumonie*, *P. mirabilis* and *S. typhimurium* were the most sensitive bacteria strains to the CQD_0 sample. After gamma irradiation, these bacteria strains were the most sensitive to the CQD_100 sample irradiated by BL. 

The unique and complex mechanisms of the antibacterial activity of CQDs involve ROS generation, degeneration of cell structure, and leakage of the cytoplasm because of DNA binding and modulation of gene expression [39]. In our investigation, all samples were shown to have antibacterial activity due to high level of singlet oxygen generation which penetrates the bacterial membrane and eradicates the bacteria, thus also confirming the photoactive properties at the same time. 

The better inhibitory effect is found among Gram positive (*S. aureus* and *MRSA*) versus Gram negative (all the other) bacteria, which is the phenomenon often presented with ROS inducing a terminal effect, probably due to different cell wall structure and permeability to CQDs [40]. Bacteria inhibition recognized by rupture of the cell due to loss of cell integrity and change in morphology, with visible cell debris, can be visually seen in AFM images as well. 

To translate gamma-ray-modified CQDs to clinical use, besides their structure, chemical composition and photoluminescent properties, very important parameters which affect their future potentials are their biocompatibilty, biodistribution and toxicity [41]. Various pieces of research, including ours, have showed that CQDs do not demonstrate any significant dark cytotoxicity [18]. Apart from cytotoxicity, one of the important biocompatibility parameters is cell proliferation, which defines a cell’s ability to attach, grow and proliferate on surfaces [42]. We found earlier that polyurethane films encapsulated by CQDs showed resistivity to adherence and proliferation by eukaryotic cells. This fact is very valuable for the potential usage of these composite films as catheters. 

Investigation of different types of CQDs (prepared by various procedures, by using different precursors and solvents, modified by doping with heteroatoms or by gamma rays or ions) will lead to great benefits and opportunities in medicine, i.e., new development of diagnostic tools, antibacterial drugs or other therapeutic drugs. In this way, human health and life quality can be improved significantly [43]. However, an understanding of physico-chemical and biological properties of CQDs is mandatory for successful application of this material in different fields of biomedicine.

## 4. Materials and Methods

### 4.1. CQDs Synthesis and Characterization

Citric acid (10 g, Sigma Aldrich, Schnelldorf, Germany) was heated in air at 210 °C for 1 h in a porcelain boat [35]. The produced soot was dissolved in water (50 mL) ultrasonically. Luminescent CQDs were filtered through 100 nm ano-disc inorganic filter membrane (Whatman, Florham Park, NJ, USA) and dialyzed for 24 h. The pH value of the CQD colloid was tuned to 7 and the concentration was adjusted to 1 mg/mL. 

The prepared CQDs were exposed to gamma irradiation at different doses (25, 50, 100 and 200 kGy) at the Institute of Nuclear Sciences Vinča (^60^Co gamma irradiation facility). All samples were characterized by different techniques: TEM, AFM, XPS, FTIR, Raman spectroscopy, PL, EPR, antibacterial and cytotoxic testing, and bioimaging.

TEM imaging was performed on a JEOL JEM-1400 operated at 120 kV. CQD samples were deposited on graphene oxide support film on lacey carbon copper grid with 400 mesh by the drop-casting method. 

A Quesant (Agoura Hills, CA, USA) microscope was used for AFM measurements operating in tapping mode in the air at room temperature with standard silicon tips from NanoAndMore GmbH (Wetzlar, Germany) with the force constant of 40 N/m. As a probe, the Q-WM300, a rotated monolithic silicon probe for non-contact high-frequency applications was used. Water dispersion of CQDs which had been gamma irradiated with doses of 25, 50, 100 and 200 kGy was deposited on mica substrate by spin coating at a spin rate of 3500 rpm for 60 s, and imaged. Gwyddion software was used for AFM image analysis [44]. 

The XPS analyses of all samples were carried out on the PHI-TFA XPS spectrometer produced by Physical Electronics Inc. (Chanhassen, MN, USA) and equipped with an Al-monochromatic X-ray source. The analyzed area was 0.4 mm in diameter and the analyzed depth was about 3–5 nm. The accuracy of binding energies was about ±0.5 eV. The C 1s spectrum was aligned to 284.8 eV. XPS spectra were fitted with Gauss–Lorentz functions and the Shirley function was used to remove the background [24,25]. Three XPS measurements were performed for every sample at different places in the irradiated area. Uncertainty of chemical composition was calculated in terms of standard deviation, which was typically 0.2–1.7% of the reported elemental concentration given in at.%.

FTIR measurements were conducted on a Perkin Elmer Spectrum Two model. The instrument operated in the ATR mode. A drop-casting method was used to deposit CQD samples on silicon substrates. Spectra were recorded at ambient temperature in the range of 400 to 4000 cm^−1^. The spectral resolution was 4 cm^−1^. 

Raman spectra of all CQD samples were recorded on a DXR Raman Microscope (Thermo Scientific, Waltham, MA, USA) by 780 nm laser. The laser power level was 10 mW. The spectra were recorded in air at room temperature with 20 s exposure time. The samples were deposited on silicon substrates.

UV-Vis spectra were recorded on a LLG-UNISPEC2 spectrophotometer (LLG international GmbH, Meckenheim, Germany) in the range of 190 to 900 nm at ambient temperature. The PL spectra of the CQD_25 and CQD_200 samples were recorded on a Fluorolog spectrofluorometer (Horiba, Kyoto, Japan). UV-Vis and PL measurements were conducted in air at room temperature. All QYs were determined using quinine sulfate (99+%, Fluka chemie GmbH, Buchs, Switzerland) in 0.1 M H_2_SO_4_ as a spectroscopic standard taking the QY of quinine sulfate in 0.1 M H_2_SO_4_ 0.58 [45].

EPR measurements were performed on a Spectrometer MiniScope 300, Magnettech, Berlin, Germany. The CQD samples were irradiated by AL and BL (3W, 470 nm) for 12 h. TEMP was used as a spin trap. With ^1^O_2_, TEMP molecules quickly react and form the stable, EPR-active product TEMP-^1^O_2_ (TEMPO). The samples in the concentration of 0.2 wt% were mixed with TEMP solution in ethanol (50% colloid, 50% ethanol solution), at a concentration of 30 mM.

ABDA (Merck, Burlington, MA, USA) were used as received. Milli-Q water (>18.2 MΩ cm) was prepared freshly before the experiment. A stock solution of ABDA in dimethyl sulfoxide (DMSO, Sigma Aldrich, Germany) of ~10 mM was used to prepare solutions of ABDA in H_2_O with 1% (*v*/*v*) DMSO. To obtain an ABDA absorbance near 1, the concentration of ~0.1 mM was chosen [34]. A mixture of 200 mL of CQDs solution and 3 mL of ABDA solution was made. For irradiation experiments, the solutions were placed into a 10 mm path-length cuvette. The intensity of the peak at 400 nm was not changed. Then, CQD samples were irradiated with blue light at 470 nm, 3W. At regular intervals (60, 120, 180 and 240 min), irradiation was stopped and the absorbance spectra were recorded on a LLG-UNISPEC2 spectrophotometer (LLG, Germany). All irradiation experiments were repeated at least three times.

### 4.2. Antibacterial Activity

The antibacterial potential of CQD_25, CQD_50, CQD_100 and CQD_200 colloids was tested against the Gram-positive bacteria *S. aureus* (ATCC 25923), *Methicillin-resistant S. aureus* (ATCC43300), and the Gram–negative bacteria *E. coli* (ATCC 25922), *K. pneumonie* (ATCC 700603), *P. mirabilis* (ATCC 29906) and *P. aeruginosa* (ATCC 27853). The bacterial strains were cultured on cation-adjusted Mueller Hinton Agar (MHA, Mueller Hinton II agar, Becton, Dickinson and Company, Franklin Lakes, NJ, USA) at 37 °C for 24 h prior to the analysis. The antimicrobial activity was evaluated regarding the ISO 20776-1 method where applicable. In brief: The F-bottom microtitre plates (Sarstedt, Numbrecht, Germany) were prepared by filling with 100 μL cation-adjusted Mueller Hinton Broth (MHB, Mueller Hinton II agar, Becton, Dickinson and Company, USA) in all but the first column (control of material sterility). The aliquot of 100 μL of each material was filled and two-fold titrated from in a further 11 steps (concentrations from 0.5 to 0.0005 mg/mL), leaving the last column serving as a bacterial growth control. Bacterial inoculums were prepared from overnight bacterial cultures suspended in sterile saline and 5 μL of bacterial suspension was added to each well, giving the final cell concentration of 5 × 10^5^ CFU/mL (range 2 × 10^5^ CFU/mL to 8 × 10^5^ CFU/mL). Immediately after preparing the inoculum, the plates were exposed to blue light (470 nm, 3 W, overnight) at a 20 cm high distance in the chamber with the temperature set at 37 °C. Plates without illumination were only exposed to ambient light and incubated in the same manner. After incubation, the plates’ extinctions were read on a spectrophotometer plate ELISA reader at 600 nm. Extinction data were used for further analyses. For statistical data analyses, the Prism 8 (GraphPad, La Jolla, CA, USA) was used. Since not all data sets conformed to a normality test (Kolmogorov-Smirnov test *p* < 0.05), nonparametric paired-sample Friedman and Dunn’s posthoc tests were used to estimate differences in the effects of samples exposed to blue light and control groups.

### 4.3. Morphology of Bacterial Strains

Bacterial cells for AFM measurements were fixed with 2.5% (*v*/*v*) glutaraldehyde solution (pH 6.8) for 2.5 h at 4 °C and washed in PBS. The morphology of the bacterial strains was recorded by AFM (Quesant, Agoura Hills, CA, USA) operating in tapping mode in the air at room temperature with standard silicon tips (NanoAndMore GmbH, Wetzlar, Germany) with the force constant of 40 N/m. As a probe, the Q-WM300, a rotated monolithic silicon probe for non-contact high-frequency applications, was used. Gwyddion software was used for AFM image analysis [44]. 

### 4.4. Cytotoxicity Assay

Cytotoxicity of CQD_25 and CQD_200 was investigated by measuring the viability of MRC-5 cells after 24 h of treatment with tested samples. The CQD_25 and CQD_200 samples were diluted in cell growth medium to final concentrations of 1 µg/mL, 10 µg/mL, 25 µg/mL, 50 µg/mL, 75 µg/mL and 100 µg/mL. MRC-5 cells were seeded in a 96-well plate at a concentration of 5 × 10^3^ cells per well and incubated overnight to allow the attachment of the cells to the bottom of the well. The next day, cells were treated with the increasing concentrations of CQD_25 or CQD_200. After 24 h of treatment, cell viability was measured using MTT assay. Absorbance was measured at 540 nm on a Tecan Infinite 200 PRO microplate reader (Tecan Group, Männedorf, Switzerland). Cell viability was presented as a percentage of the control, which was set to 100%. The control for cells treated with CQD_25 and CQD_200 was MRC-5 cells treated with vehicle control.

### 4.5. Fluorescence Microscopy

MRC-5 cells (obtained from the ATCC culture collection) were grown in high-glucose Dubelco’s Modified Eagle’s Medium (DMEM) supplemented with 10% fetal bovine serum and penicillin/streptomycin solution (10,000 units penicillin and 10 mg/mL streptomycin) (all from Thermo Fisher Scientific) and maintained in a humidified incubator at 37 °C with 5% CO_2_. MRC-5 cells were seeded on cover slips in 12-well plates at a density of 4 × 10^4^ and incubated overnight. The next day, cells were treated with medium containing 200 µg/mL solution of CQD_25 or CQD_200 (deionized water was used as a vehicle control). After 48 h of treatment, cells were washed three times in PBS and visualized using a Spectrum Aqua filter on an Olympus BX51 fluorescence microscope (Olympus, Tokyo, Japan). Images were acquired with a 20× objective and analyzed with Cytovision 3.1 software (Applied Imaging Corporation, Santa Clara, CA, USA).

## 5. Conclusions

Regarding the above results, there is proven antibacterial activity of CQDs irradiated by gamma rays at different doses. Structural analyses conducted by TEM and AFM showed that CQDs had a spherical-like shape with irradiation-dose-dependent average diameters and heights. By increasing the irradiation dose, the content of hydroxyl groups was increased, as shown by XPS and FTIR analyses. The PL of all samples showed PL excitation–emission dependence and blue light emission. When comparing ambient and blue light conditions, the sample CQD_100 showed antibacterial effect against all tested strains with a significantly better antibacterial effect when samples were exposed to blue light (*p* < 0.0001). All tested samples had a viability range tat was within 89–101% for all concentrations of all tested samples, concluding that there was no cytotoxicity. Cellular uptake of CQD_25 and CQD_200 samples was better compared with the control sample.

In this way, we verify that gamma rays induced structural modification of CQDs affect their antibacterial activity, i.e., this improves their antibacterial effect and indicates their usage in potential antibacterial products, especially against *S. aureus* and *MRSA*.

## Figures and Tables

**Figure 1 antibiotics-12-00919-f001:**
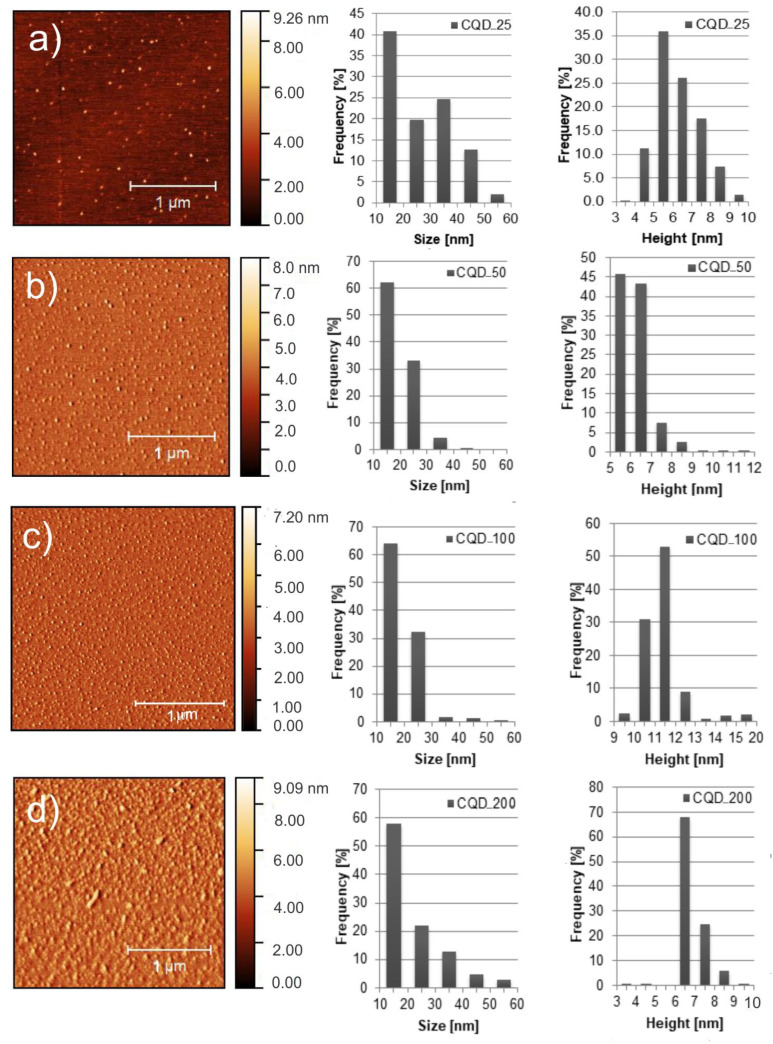
Top view AFM images including particle size and height distributions of: (**a**) CQD_25, (**b**) CQD_50, (**c**) CQD_100 and (**d**) CQD_200 samples.

**Figure 2 antibiotics-12-00919-f002:**
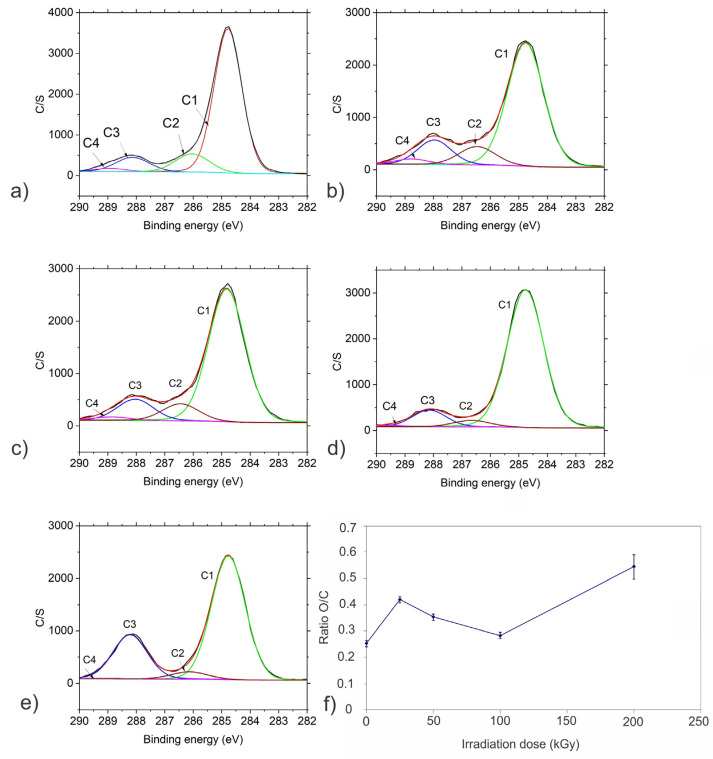
Fitted XPS spectra of (**a**) CQD_0, (**b**) CQD_25, (**c**) CQD_50, (**d**) CQD_100, (**e**) CQD_200 samples, (**f)** O/C ratio on sample surface as a function of irradiation dose. Peak C1 presents C-C/C-H bonds, peak C2 presents C-O/C-OH/C-O-C bonds, peak C3 presents C=O/O-C-O/CO3 bonds and peak C4 presents O=C-O bonds.

**Figure 3 antibiotics-12-00919-f003:**
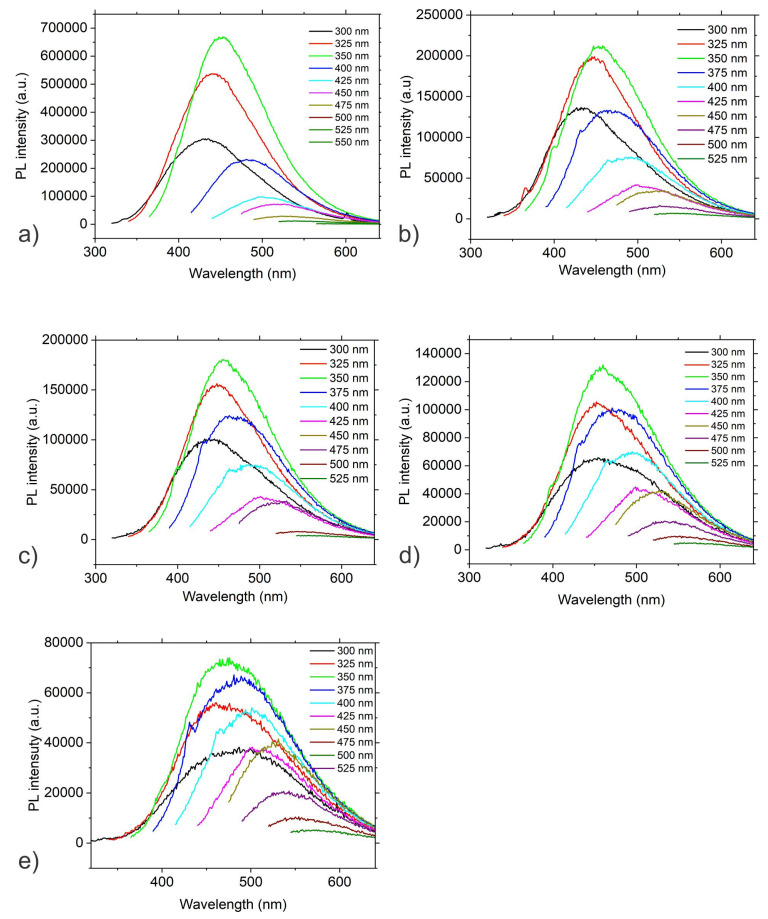
PL spectra of (**a**) CQD_0, (**b**) CQD_25, (**c**) CQD_50, (**d**) CQD_100 and (**e**) CQD_200 samples.

**Figure 4 antibiotics-12-00919-f004:**
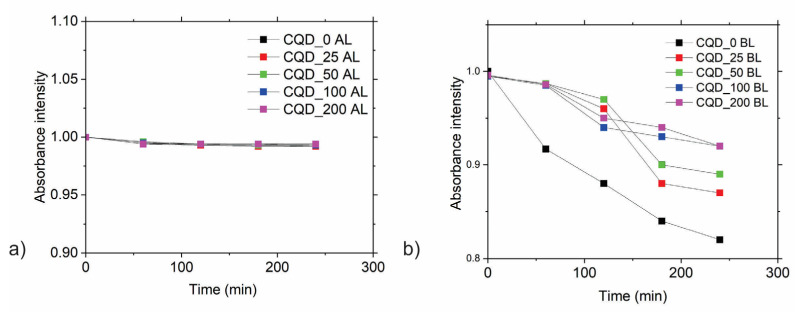
Photobleaching of ADBA under (**a**) ambient light (AL) and (**b**) blue light (BL) in the presence of all samples: CQD_0, CQD_25, CQD_50, CQD_100, CQD_200. All absorbance spectra of ABDA were recorded at 398 nm, normalized at the start of the irradiation, and averaged over several repeat experiments at a similar concentration of CQDs. Standard deviations for each measurement were smaller than the size of the symbols.

**Figure 5 antibiotics-12-00919-f005:**
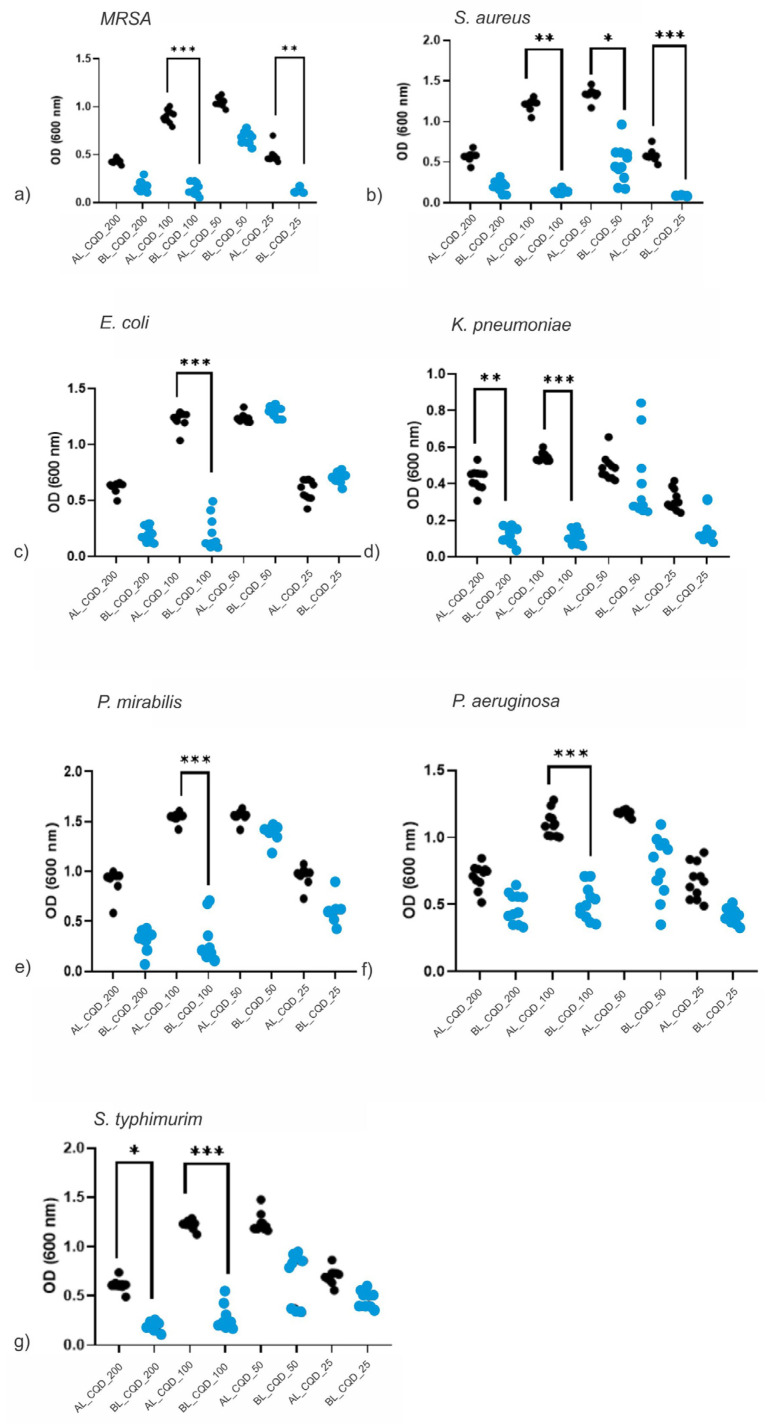
The bacterial growth change under CQDs compared between BL (blue dots) and AL (black dots) conditions. All tested bacteria, (**a**) MRSA, (**b**) *S. aureus*, (**c**) *E. coli*, (**d**) *K. pneumonie*, (**e**) *P. mirabilis*, (**f**) *P. aeruginosa* and (**g**) *S. typhimurim*, revealed some reduction in bacterial density under BL, but with CQD_100, all reductions were statistically significant. Note: as lower OD value means less bacterial growth and better antibacterial activity. Asterisks represent significant differences (* *p* < 0.05, ** *p* < 0.01, *** *p* < 0.001).

**Figure 6 antibiotics-12-00919-f006:**
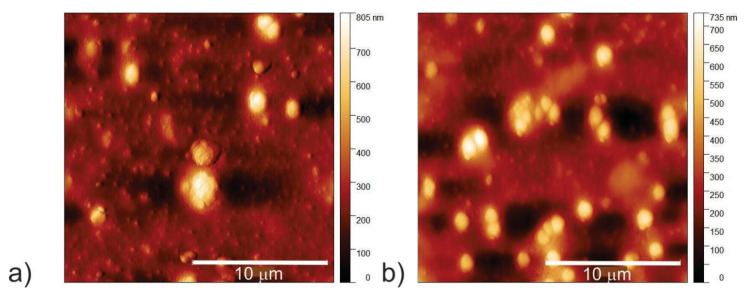
Top view AFM images of MRSA bacterial strains treated by CQD_100 samples under (**a**) AL and (**b**) BL irradiation.

**Figure 7 antibiotics-12-00919-f007:**
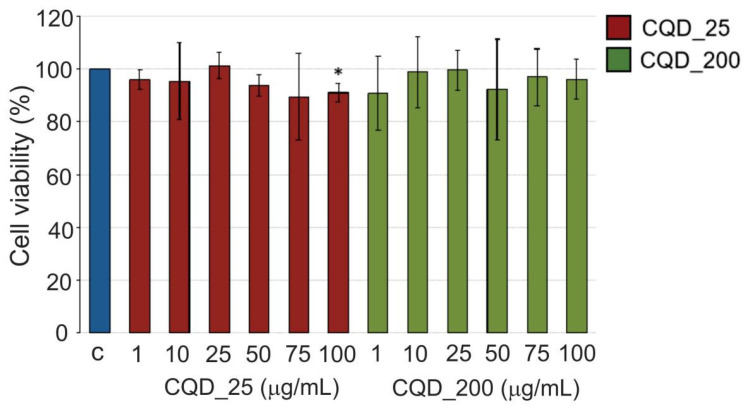
Cytotoxity of CQD_25 and CQD_200 samples against MRC-5 cells. MRC-5 cells were treated for 24 h with increasing concentrations (1, 10, 25, 50, 75 and 100 µg/mL) of two tested CQD samples. Cell viability was expressed as the percentage of absorbance relative to the vehicle-control-treated MRC-5 cells for CQDs samples. Data are presented as the mean ± SD of at least three independent experiments. Asterisks indicate statistical significance (* *p* < 0.05). SD—standard deviation.

**Figure 8 antibiotics-12-00919-f008:**

Cellular uptake of CQD_25 or CQD_200. Representative fluorescence images of (**a**) control MRC-5 cells (treated with vehicle control) and cells treated with 200 μg/mL of (**b**) CQD_25 and (**c**) CQD_200 for 48 h. Images were acquired at 20× magnification. Scale bar is 100 µm. Red arrows indicate MRC-5 cells with internalized irradiated CQDs samples.

**Table 1 antibiotics-12-00919-t001:** Surface composition in at.% from XPS data.

Dose (kGy)	C (at.%)	O (at.%)	N (at.%)
0	77.6 ± 0.7	19.0 ± 0.7	3.4 ± 0.5
25	71.6 ± 0.4	27.5 ± 0.6	1.0 ± 0.2
50	74.8 ± 0.5	24.6 ± 0.7	0.6 ± 0.2
100	79.1 ± 0.7	20.4 ± 0.7	0.5 ± 0.2
200	68.9 ± 1.7	31.2 ± 1.7	0.0

**Table 2 antibiotics-12-00919-t002:** Raman spectra of all samples.

G Band (cm^−1^)	CQD_0	CQD_25	CQD_50	CQD_100	CQD_200
G11	1580.2			1575.7	1575.6
G12		1590.6	1593.6	1591.6	1591.5
G2	1602.5	1614.3	1612.3	1606.2	1610.3
FWHM1	12.7			18.0	19.7
FWHM2		25.1	30.4	20.8	26.8
FWHM3	44.7		17.2	43.2	27.6
I_D_/I_G_	1.6	1.5	1.9	1.8	2.1

**Table 3 antibiotics-12-00919-t003:** Fitted PL spectra of all samples. The excitation wavelength was 400 nm.

Sample	P1 (nm)	P (nm)	P2 (nm)	S1/S2
CQD_0	472.29	484.65	519.89	1.37
CQD_25	473.55	487.31	521.88	1.09
CQD_50	474.48	489.69	523.61	1.07
CQD_100	477.42	495.89	528.08	1.09
CQD_200	483.26	498.46	536.68	1.18

**Table 4 antibiotics-12-00919-t004:** QY of CQD_0, CQD_25, CQD_50, CQD_100 and CQD_200 samples.

Sample	λ_exc_ (nm)	QY
CQD_0	440	0.738
CQD_25	440	0.250
CQD_50	440	0.199
CQD_100	440	0.148
CQD_200	440	0.081

## Data Availability

Not applicable.

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
