# Peer review of "Employing Gamma-Ray-Modified Carbon Quantum Dots to Combat a Wide Range of Bacteria"

_antibiotics, 2023, doi:10.3390/antibiotics12050919_

Round 1

Reviewer 1 Report

The authors obtained gamma rays modified carbon quantum dots and characterized them using spectroscopic tools and microscopy. The effect of these compounds was investigated on many bacterial strains. There are some points to be improved and better explained before the manuscript publication and some of them are listed below:

(1) Many abbreviations were not defined the first time they appear. It is very difficult to read the article due to this point.

(2) this conclusion “It is obvious from Figure 1 that diameters of CQDs do not depend on the applied irradiation” is based on a visual analysis without statistical test.

(3) is unclear if the experiments about chemical composition were performed in triplicate. Are missing the standard deviation. Repetition is necessary for statistical analysis Figure 2f shows data without standard deviation. Are usually chemical quantifications to be performed in triplicate. Without the information about the method variability is not possible to conclude about the differences. The experiment repetition is essential to the assessment of the variability linked with the analytical methods.

(4) the word “obvious” appears 7 times and is not usual. Many times this word is not consistent with the data described.

(4) the x axis of figure S3 is inverted

(5) when the results are separated from the discussion, they must not contain references. Only the results description is expected in the results section. Was not found a section with the title “discussion”. The number of subtitles must be revised, the methodology is not title number 2 in MDPI journals.

(5) the authors must discuss what are the challenges involved with pre-clinical and clinical trials using gamma rays modified carbon quantum dots. Information about their toxicity and drug-like properties, water solubility must be available to readers. The toxicity is poorly discussed.

(6) the manuscript contains 5% of similarity with previous publications of the authors (https://doi.org/10.21203/rs.3.rs-2419272/v1). Please change the text to avoid plagiarism.

(7) The use of expressions “good” and “very good” without numerical basis occurs many times in the manuscript. Please see that the use of these constructions is so wide and unspecific.

Minor editing of the English language is required, but the structure of sentences may be improved. 

Author Response

Dear Sir,

We read referee’s comments carefully. All his suggestions were accepted and the manuscript was corrected according his remarks.

Please find below the answers to referee’s comments:

Review 1

  • Many abbreviations were not defined the first time they appear. It is very difficult to read the article due to this point.

We defined abbreviations the first time they appeared as referee suggested.

  • this conclusion “It is obvious from Figure 1 that diameters of CQDs do not depend on the applied irradiation” is based on a visual analysis without statistical test.

We omitted word “obvious” and rephrased the appropriate text completely. Statistical test was conducted by Gwyddion software. We analysed about 20 images with more than 500 nanoparticles. Pages 2-3

  • is unclear if the experiments about chemical composition were performed in triplicate. Are missing the standard deviation. Repetition is necessary for statistical analysis Figure 2f shows data without standard deviation. Are usually chemical quantifications to be performed in triplicate. Without the information about the method variability is not possible to conclude about the differences. The experiment repetition is essential to the assessment of the variability linked with the analytical methods.

Experiments about chemical composition were performed in triplicate. Standard deviations were added in Table 1 and Figure 2f. pages 4 and 5.

  • the word “obvious” appears 7 times and is not usual. Many times this word is not consistent with the data described.

All “obvious” words were removed from the manuscript and the text was rephrased.

  • the x axis of figure S3 is inverted

The x axis was changed as referee’s requested.  Supporting information page 5 new Figure S4

  • when the results are separated from the discussion, they must not contain references. Only the results description is expected in the results section. Was not found a section with the title “discussion”. The number of subtitles must be revised, the methodology is not title number 2 in MDPI journals.

The section “3.Discussion” (page 12) was inserted in the manuscript. The methodology section was designated with 2. Sorry for this unintentional mistake.

  • the authors must discuss what are the challenges involved with pre-clinical and clinical trials using gamma rays modified carbon quantum dots. Information about their toxicity and drug-like properties, water solubility must be available to readers. The toxicity is poorly discussed.

All these referee’s suggestions were inserted in section 3.Discussion and elaborated. Page 12

  • the manuscript contains 5% of similarity with previous publications of the authors (https://doi.org/10.21203/rs.3.rs-2419272/v1). Please change the text to avoid plagiarism.

The text was changed as requested.

  • The use of expressions “good” and “very good” without numerical basis occurs many times in the manuscript. Please see that the use of these constructions is so wide and unspecific.

Expressions “good” and “very good” were omitted from text and the text rephrased.

Best regards,

Biljana Todorovic Markovic

Reviewer 2 Report

This manuscript is based on spectroscopy analysis of Carbon Quantum Dots samples that have been irradiated with several doses of gamma rays. Moreover, TEM and AFM images were analysed to verify the surface morphology after irradiation doses and also the antibacterial and cytotoxic activities from seven different bacteria were researched. I can recommend its publication after minor revisions.           

1)      Surface Morphology: Based on TEM and AFM images, the authors have concluded that the diameter of CQDs is independent of applied irradiation doses (Page 2 – Lines 89-100), although the dispersion of size and height of CQDs decreased while irradiation doses were increasing (Figure 1). Would you mind rewriting this sentence/explaining me why exactly you did not consider the doses influencing the distribution of size and height of CQDs?

2)      Chemical composition: How many samples were investigated? Have all samples ever presented the same percentual composition related to Carbon, Oxygen and Nitrogen after each irradiation dose? Certainly, this information will be able to clarify if the Nitrogen is being adsorbed by the sample surface (Page 3, Lines 115-116).

3)   Photoluminescence spectra: What were the quantum efficiency of the CQDs samples?

I suggest reviewing the typos along the text, e.g. (Page 2, Line 56), (Page 6, Line 217) and also removing the repetitive phrase "is obvious" during the analysis of the Results. Vide first question above (Comments and suggestions for Authors), although along the manuscript you have written that the analysis is obvious. Moreover, if the analysis is obvious, you don't need to analyse.

Author Response

Dear Sir,

We read all your suggestions carefully and revised manuscript as requested. Please find below the answers to referee’s remarks:

Report review 2

  • Surface Morphology: Based on TEM and AFM images, the authors have concluded that the diameter of CQDs is independent of applied irradiation doses (Page 2 – Lines 89-100), although the dispersion of size and height of CQDs decreased while irradiation doses were increasing (Figure 1). Would you mind rewriting this sentence/explaining me why exactly you did not consider the doses influencing the distribution of size and height of CQDs?

We analyzed all recorded AFM images once again. We got the similar results related to particle diameters and heights. But we determined the average values of these parameters and conclusion the following: there was decreasing trend of average particle diameter to irradiation dose of 200 kGy. At dose of 200 kGy average particle diameter started to increase. In the case of average height, it increased to dose of 200 kGy and at a dose of 200 kGy it decreased. The manuscript text was revised according to these observations. TEM micrographs were transferred to Supporting information as Figure S1. Pages 2 and 3

  • Chemical composition: How many samples were investigated? Have all samples ever presented the same percentual composition related to Carbon, Oxygen and Nitrogen after each irradiation dose? Certainly, this information will be able to clarify if the Nitrogen is being adsorbed by the sample surface (Page 3, Lines 115-116).

Five samples were investigated and each of them was investigated in triplicate. The concentration of each element and appropriate error are presented in Table 1. Page 4. As written in the section 4.1,

Three XPS measurements were performed for every sample at different places of irradiated area. Uncertainty of chemical composition was calculated in terms of standard deviation, which was typically 0.2 – 1.7 % of reported elemental concentration given in at.%. pages 13 and 14

  • Photoluminescence spectra: What were the quantum efficiency of the CQDs samples?

 We determined QY of all CQDs samples (Table 4, page 7, section 4.1, page 14)

  1. Comments on the Quality of English Language

I suggest reviewing the typos along the text, e.g. (Page 2, Line 56), (Page 6, Line 217) and also removing the repetitive phrase "is obvious" during the analysis of the Results. Vide first question above (Comments and suggestions for Authors), although along the manuscript you have written that the analysis is obvious. Moreover, if the analysis is obvious, you don't need to analyse.

We corrected the text as requested.

Best regards

Biljana Todorovic Markovic

Reviewer 3 Report

The intention of this study is good but I cannot accept it in its current form. I have some observations.

In the title, “ Employing” please check whether it is necessary to use.  

Abstract: Please start your abstract with the aims of the study, not the results. Conclude the abstract with a brief summary of your results. Rewrite abstract

Introduction: lines 48,49 please use references for individual materials

Line 68 Please follow reference order, reference 10-14 are missing

Line 78, this is not the first study, similar studies can be found in the search engine, please revise

Line 89. The first appearance of TEM and AFM should be in full form, please check all abbreviations throughout the manuscripts.

Figure 1: TEM images are not clearly defined. AFM images should be more magnified and identical marks should be placed on images and discussed in the main manuscripts as well as figure ligands.

Figure 2: Please add XPS wide scan first then, magnification, and give a brief explanation in figure ligands.

In the results, please show Raman spectra and proper citation of your figures in the main manuscripts.

Figure 3, Please make it readable

Line 181-182, justify your results

Figure 5, please use colors, and explain it in figure legends.  

Please use clear images for Figure 6

Figure 8, identify your findings

If authors write results and discussion combinedly, there must have consistency. I cannot see any consistency.

If materials and methods are in heading 4, Why the subheading is started with  2.1? it is a carelessness

Rearrange the manuscripts, and make consistency with results and discussion.

Reflect your aims of the study in the conclusion

Please check similarity

Please check English, especially, sentence making.

Supplementary data and their citation should be clearly defined.

Please add appropriate figure ligands for all supplementary figures or explain them in the supplementary text file.

I hope these comments are useful to improve the manuscripts. I look forward to seeing the revised version.

Please check English, especially, sentence making.

Author Response

Dear Sir,

We read all your suggestions carefully and revised manuscript as requested. Please find below the answers to referee’s remarks:

Report review 3

1.Abstract: Please start your abstract with the aims of the study, not the results. Conclude the abstract with a brief summary of your results. Rewrite abstract

We revised the abstract as requested.

  1. Introduction: lines 48,49 please use references for individual materials

We inserted the appropriate references (ref.1-3)

  1. Line 68 Please follow reference order, reference 10-14 are missing

There was mistake in the manuscript. This mistake was corrected, ref. 12-18

  1. Line 78, this is not the first study, similar studies can be found in the search engine, please revise

We rephrased the sentence (page 2)

  1. Line 89. The first appearance of TEM and AFM should be in full form, please check all abbreviations throughout the manuscripts.

We corrected this remark and put the full name of abbreviation in the first appearance.

6.Figure 1: TEM images are not clearly defined. AFM images should be more magnified and identical marks should be placed on images and discussed in the main manuscripts as well as figure ligands.

We separated TEM and AFM images (Figure S1 and Figure 1). AFM images were clarified and we put the same marks for sample designations.

Figure 2: Please add XPS wide scan first then, magnification, and give a brief explanation in figure ligands.

We added wide XPS scan of CQD_0 sample (Figure S2a) and put brief explanation in figure caption (Figure 2 and Figure S2).

  1. In the results, please show Raman spectra and proper citation of your figures in the main manuscripts.

We put Raman spectra in Supporting information-Figure S5 and corrected the citation of figures in the text.

  1. Figure 3, Please make it readable

We enlarged the Figure 3.

  1. Line 181-182, justify your results

We clarified our results section 3.Discussion.

  1. Figure 5, please use colors, and explain it in figure legends.  

We used blue colour for samples tested under blue light. We put new figure caption for Figure 5 and appropriate explanations.

  1. Please use clear images for Figure 6

We used the best images for Figure 6 which we recorded.

  1. Figure 8, identify your findings

We identified our findings in Figure 8 by red arrows.

  1. If authors write results and discussion combinedly, there must have consistency. I cannot see any consistency.

We wrote the new section 3.Discussion in which we elaborated all our findings.

  1. If materials and methods are in heading 4, Why the subheading is started with  2.1? it is a carelessness

Sorry for this unintentional mistake. We corrected the designation of methodology sections.

  1. Rearrange the manuscripts, and make consistency with results and discussion.

We rearranged the manuscript and inserted new section 3.Discussion

  1. Reflect your aims of the study in the conclusion

We corrected the aims in the conclusion.

  1. Please check similarity

We checked the similarity as requested.

  1. Please check English, especially, sentence making.

We tried to improve English.

  1. Supplementary data and their citation should be clearly defined.

We corrected the mistakes.

  1. Please add appropriate figure ligands for all supplementary figures or explain them in the supplementary text file.

We corrected supporting information file as requested.

  1. I hope these comments are useful to improve the manuscripts. I look forward to seeing the revised version.

Thank you for careful reading of our manuscript and the appropriate comments.

Best regards

Biljana Todorovic Markovic

Round 2

Reviewer 1 Report

Thank you for having given me the opportunity to review again the manuscript entitled “Employing gamma rays modified carbon quantum dots to combat wide range of bacteria”. I am satisfied with the answers of the author. Thus, in my opinion, the manuscript meets the requirements for publication in Antibiotics, and I recommend accept it in its current form. 

Minor editing of English language required

Reviewer 3 Report

The authors have answered mostly. I have no more comments.